# Quality Evaluation of Shiitake Blanched and Centrifuged Broths as Functional Instant Drinks

**DOI:** 10.3390/foods12152925

**Published:** 2023-08-01

**Authors:** Shin-Yu Chen, Jim Tseng, Cheng-Rong Wu, Sheng-Dun Lin

**Affiliations:** 1Department of Food Science, National Pingtung University of Science and Technology, Pingtung 912301, Taiwan; sychen@mail.npust.edu.tw; 2Department of Food Science and Technology, Hungkuang University, 1018, Sec. 6, Taiwan Boulevard, Shalu District, Taichung 433304, Taiwan; tsengxjim85@gmail.com (J.T.); sam2763366@gmail.com (C.-R.W.)

**Keywords:** shiitake, blanched broth, centrifuged broth, indigestible dextrin, sensory evaluation

## Abstract

In the process of making mushrooms into vacuum-fried crisps, the resulting blanched broth (BB) and centrifuged broth (CB) are often discarded, thereby increasing the amount of wastewater and treatment costs. This study measured the proximate compositions, bioactive components, taste components, and minerals of freeze-dried BB and CB and then used functional indigestible dextrin (Fibersol-2) as a carrier to make these two broths into instant drinks. The solids of the BB and CB contained protein (16.88–19.21%), fat (0.01–0.23%), ash (12.89–13.50%), carbohydrate (67.28–70.00%), sugars and polyols (40.55–45.68%), free amino acids (6.58–6.69%), 5′-nucleotides (0.98–1.47%), and bioactive components, especially polysaccharides (4.53–7.45%), ergothioneine (both 0.19%), and total phenols (0.15–0.36%). The equivalent umami concentration of BB was 2.77-fold higher than that of the CB. Both BB and CB showed compositions and essential minerals that are rich in taste. Using a nine-point hedonic test, it was found that the solid contents of BB and CB in the instant drink affected the consumer’s preference. The flavor and overall preference of instant drinks with 2.5% BB or CB were the best amongst consumers. Overall, the BB and CB were rich in nutrients and bioactive and taste components and could be developed as a functional food in the form of a drink.

## 1. Introduction

*Lentinula edodes* (Berk.) Pegler, or shiitake in Japanese and xianggu in Chinese, is an edible and medicinal mushroom native to East Asia. It is the second most widely cultivated mushroom around the world because of its unique flavor, texture, and medicinal and nutritional value [1,2,3,4]. Shiitake is rich in polysaccharides, proteins, flavor compounds, amino acids, sterols, dietary fibers and vitamins, and is low in fat and sodium [2,5,6]. The protein, ash, and carbohydrate of fresh shiitake cap and stipe are 15.5–18.5%, 4.3–6.2%, and 72.3–77.6%, respectively [5]. In Taiwan, fresh shiitake is divided into special grade, premium grade, and excellent grade, and in three different size classes according to the diameter of the caps: small (3–5 cm), medium (5–7 cm), and large (>7 cm) [5]. Fresh small-sized shiitake of good grade are very perishable, and they have a low commodity value in the market. Therefore, they are commonly used as raw materials for processing mushrooms, such as vacuum-fried crisps and prepared foods [3,5,7,8]. Nevertheless, a few studies have utilized shiitake as an ingredient to produce alcoholic and non-alcoholic drinks by fermentation [9,10,11].

Hot water blanching is one of the most common traditional pretreatment methods for vegetable processing and is often used to reduce or eliminate the bitterness of vegetables. The main purpose of blanching shiitake is to destroy enzyme activity and reduce the deterioration rate [5]. Blanching is also an important step in the process of preparing shiitake as vacuum-fried crisps and prepared foods. It can improve the color of the vacuum-fried samples and reduce oil absorption [7]. After hot water blanching, water is retained in the shiitake, and the moisture level is increased. Then, centrifugation is utilized to remove some of the moisture from the mushrooms, so that the mushrooms can absorb the seasoning liquid more easily [3,8]. 

However, the blanching process requires large amounts of water and reduces the nutrients in the shiitake [4,12]. Most of the proximate composition, minerals, sugar and polyols, amino acids, 5′-nucleotides, and bioactive compounds of shiitake leak into the blanching water during the process, but carbohydrates remain [5]. The blanched broth (BB) and centrifuged broth (CB) produced are often discarded, resulting in increased waste water and treatment costs [8,12,13]. The BB and CB were subjected to preliminary quality analyses, and they were found to contain many taste-related and bioactive components. Therefore, finding methods for reusing processing by-products in order to increase added value and manufacturers’ profits and reduce waste water represents important research [14,15].

Indigestible dextrin is often used in the manufacturing of foods and drugs because of its many applications and benefits. Fibersol-2 (FS-2) is a highly digestion-resistant maltodextrin fiber supplement, and it is used as a bulking agent [16,17,18]. The process of making FS-2 includes the heating and enzymatic treatment of corn starch, generating a random distribution of α- and β-(1,4), (1,6), (1,2), and (1,3) linkages with an average molecular weight of 2000 Da [16]. Only α-(1,4) and (1,6) linkages in the FS-2 can be hydrolyzed and digested in the human digestive system, and they have an energy value of 0.5 kcal/g, found through a series of in vivo and in vitro experiments [17]. FS-2 has no known acute toxicity or mutagenicity [19]. FS-2 also has many good processing characteristics, such as low viscosity, water solubility, acid stability, freeze/thaw stability, and heat stability, making it suitable for use in products such as juices, sauces, puddings, and drinks. Several studies have reported that FS-2 has the following beneficial effects: improves intestinal regularity [20], moderates postprandial blood glucose levels [21], lowers serum cholesterol and triglyceride levels [22], and regulates satiety and hunger hormones to enhance satiety after a meal [23]. Therefore, it is important to explore the use of FS-2 as a carrier to make blanched and centrifuged broths into instant drinks with health benefits. Additionally, Nutriose is also a commercial resistant maltodextrin [24,25], and it may be used for further comparative study.

To the best of our knowledge, there are few studies on the quality characteristics of BB and CB, especially on their development into functional instant drinks. As a result, the purpose of this study was to utilize the BB and CB from shiitake to develop a novel functional product in the form of an instant drink, with FS-2 as the carrier. The contents of proximate compositions, sugars and polyols, free amino acids, 5′-nucleotides, bioactive components, and minerals in BB and CB were determined. The physical and hedonic sensory characteristics of instant drinks containing them were also evaluated. 

## 2. Materials and Methods

### 2.1. Materials and Reagents

Small-sized, good-grade fresh shiitake were purchased from a mushroom farm in Puli, Nantou, Taiwan. After collecting the shiitake, they were packed in polyethylene bags and then transported to Hungkuang University in a refrigerated truck (4–7 °C). The FS-2 was purchased from Matsutani Chemical Industrial Co. (Hyogo, Japan).

Gallic acid, Folin–Ciocalteu phenol reagent, sugars and polyols, amino acids and 5′-nucleotides, γ-aminobytyric acid, ergosterol, ergothioneine, sodium dodecyl sulfate, trimethylamine, betaine, 2-mercapto-1-methylimidazole, sodium acetate, tetrahydrofuran, and 1,4-dithiothreitol were purchased from Sigma-Aldrich (St. Louis, MO, USA). Minerals were purchased from AccuStandard, Inc. (New Haven, CT, USA). Ethanol (95%) was purchased from Echo Chemical (Miaoli, Taiwan). Nitric acid was purchased from J.T. Baker (Radnor, PA, USA). Hydrogen peroxide was purchased from Honeywell International (Detroit, MI, USA). Hydrochloric acid, sulfuric acid, and sodium phosphate were purchased from Union Chemical Works (Hsinchu, Taiwan). Methanol and acetonitrile were purchased from Avantor Performance Materials (Radnor, PA, USA). Hexane was purchased from Tedia (Fairfield, OH, USA). Cupric sulfate, anhydrous sodium carbonate, and sodium hydroxide were purchased from Shimakyu’s Pure Chemicals (Osaka, Japan). Phenol reagent was purchased from Wako Pure Chemical Industries (Osaka, Japan). Potassium sulfite was purchased from Nihon Shiyaku Reagent (Tokyo, Japan). Methyl red pure was purchased from Koch Light Research Laboratories (Gauteng, South Africa). Methylene blue was purchased from Katayama Chemical Industries (Osaka, Japan).

### 2.2. Blanched and Centrifuged Broths

#### 2.2.1. Preparation

After washing fresh shiitake (running water, 30 s), we made up a ratio of washed shiitake (2.5 kg) to boiling water of 1:5 (*w*/*w*) and blanched the mushrooms for 5 min. After blanching 5 batches of shiitake with the same boiling water, they were cooled and filtered with a Mini Jet Filter (5–7 mm, Buon Vino Manufacturing, Cambridge, ON, Canada), taking the filtrate as the blanched broth (BB). We immediately put the above blanched shiitake into an ice bath for 1.5 min to cool. The cooled samples were put into plastic baskets, and we placed them in a freezer at −25 °C for 24 h. The frozen shiitake were then left to defrost in the refrigerator at 4–7 °C. The thawed shiitake were centrifuged in a stainless steel centrifugal separator (TF-16, Tung Fu Machinery, Taichung, Taiwan) at 1400 rpm for 2 min. The centrifuged broth thus collected was filtered through a Mini Jet Filter, taking the filtrate as the centrifuged broth (CB). The process of preparing BB and CB was repeated 3 times. Parts of the BB and CB were subjected to physicochemical quality characteristic analyses after freeze-drying (FD). The remaining BB and CB were prepared for use in instant drinks. Shiitake is edible and treated with regular processing steps; no food safety concerns exist.

#### 2.2.2. Determination of Quality Characteristics of Blanched and Centrifuged Broths

The blanched broth yield (%) = (filtered weight of blanched broth after blanching 5 batches of washed shiitake/weight of initial blanching water) × 100. Centrifuged broth yield (%) = (weight of centrifuged broth/weight of blanched shiitake) × 100. The moisture in BB and CB was determined by American Association of Cereal Chemists Approved Methods 44-40.01 [26]. The soluble solids and pH values of BB and CB were measured with a hand refractometer (Atago Co., Tokyo, Japan) and a pH meter (MP225, Mettler-Toledo, Switzerland), respectively.

#### 2.2.3. Determination of Proximate Compositions, Sugars, and Polyols

The crude ash, crude fat, moisture, and crude protein of the freeze-dried broths were determined by the methods 08-01.01, 30-25.01, 44-40.01, and 46-11.02, respectively, as approved by the American Association of Cereal Chemists [26]. The nitrogen conversion factor of the sample used to calculate the crude protein content was 4.38 [27]. On a dry basis, the carbohydrate content (%) was calculated by subtracting the protein, fat, and ash content from 100% dry matter. The sugars and polyols of freeze-dried broths were determined according to the method of Tsai et al. [28]. We identified and quantified each of the sugars and polyols based on a calibration curve of authentic compounds.

#### 2.2.4. Determination of Free Amino Acids, 5′-Nucleotides, and Equivalent Umami Concentration

The free amino acids of the freeze-dried broths were determined according to the method used by Hou et al. [29]. The 5′-nucleotides were measured according to the method of Tsai et al. [26]. Each compound was identified and quantified using the calibration curve of the authentic compound. The equivalent umami concentration (EUC, mg sodium glutamate (MSG)/100 g) was calculated according to the addition equation established by Yamaguchi et al. [30,31]. EUC gives the concentration of MSG, which is equivalent to the relative umami intensity of a mixture of MSG and 5′-nucleotides. 

#### 2.2.5. Determination of Minerals

Minerals were determined following the method described in CNS [32]. The powdered sample (0.5 g) was added to 2 mL of 30% H_2_O_2_ and 10 mL of 65% HNO_3_. The mixture was heated on a pre-heated hot plate, and 5 mL of H_2_O_2_ was added. The following mixture was heated to 200 °C for 10 min and then cooled to room temperature. After cooling, the solution was put into a microwave digester (SINEO Microwave Chemistry Technology, Shanghai, China) to perform a digestion reaction at 700 W. After digestion, the solution was quantified to 20 mL with deionized water and filtered through a 0.22 µm syringe filter. The filtered solution was analyzed by high-resolution ICP-MS (Thermo Fisher Scientific, Waltham, MA, USA). Each mineral was identified and quantified on the basis of the calibration curve of the authentic compound.

#### 2.2.6. Determination of Bioactive Compounds

According to the method previously described [33], ergothioneine and γ-aminobutyric acid were extracted and determined, and they were quantified using the calibration curve of the corresponding authentic compound. The measurement of crude polysaccharides was performed according to the method of Mau et al. [34]. The determination of total phenols was made according to the method of Mau et al. [5]. The analysis of ergosterol was based on the method of Qian and Sheng [35]. 

### 2.3. Instant Powder and Drink

#### 2.3.1. Preparation

The BB and CB were vacuum-concentrated (70 cmHg) with an outer pot temperature of 95 ± 2 °C and an inner pot of 70 ± 2 °C in a vacuum extraction concentrator (GV-800, Amos Instruments, Taipei, Taiwan). When the solid contents of the blanched broth concentrate (BBC) or the centrifuged broth concentrate (CBC) had reached 6%, the concentrate was mixed with FS-2 in a dry matter ratio of 1:3, 1:4, or 1:5, and then, the solid content of the mixed solution was adjusted to 20% with water. The mixed solution was spray-dried in a spray-dryer (SD-1000, Eyela, Tokyo Rikakikai, Tokyo, Japan). The working conditions for the spray-drying were an inlet temperature of 180 °C, an outlet temperature of 85–90 °C, a dry air volume of 0.65 m^3^/min, a spraying air pressure of 250 kPa, and a pumping volume of 4.2 mL/min. After being spray-dried, the instant drink powder was packed in aluminum foil laminated bags (PET/Al/PE), and ratios of 1:3, 1:4, and 1:5 were made up, designated as SD13, SD14, and SD15, respectively. 

In terms of brewing conditions, the instant powders were brewed with boiling water into drinks containing 1.0%, 1.5%, 2.0%, 2.5%, 3.0%, and 3.5% of dried broth, designated as 1.0, 1.5, 2.0, 2.5, 3.0, and 3.5 instant drinks, respectively.

#### 2.3.2. Determination of Physical Quality Characteristics

The moisture of the instant powder was determined according to method 44-40.01 of AACC [20]. The water activity was determined using an AquaLab CX-2 water activity meter (Decagon Devices, Pullman, WA). The reflective color of the powder and the transmittance color of the drink were determined using a spectrophotometer (CM-5, Konica Minolta, Tokyo, Japan). The spectrophotometer was set for CIE *L**, *a**, and *b** values with a D65 illuminant at 10°. The hue angle (*h**, arctan (*b*/a**), first quadrant (*+a**, *+b**); 180 + arctan (*b*/a**), second quadrant (*−a**, *+b**)) [36,37], chroma (*c**, $\sqrt{{a*}^{2}+{b*}^{2}}$), and total color difference (Δ*E*) values were calculated from the *L**, *a** and *b** values. The Δ*E* of the powder and drink in comparison to the control was calculated using the equation Δ*E* = $\sqrt{(L_{c}*-L_{s}*{)}^{2}+{\;(a_{c}*-a_{s}*)}^{2}+{\left(b_{c}*-b_{s}*\right)}^{2}}$, where *L**, *a*,* and *b** are the color coordinates in the control (c) and the powder or drink (s). Each analysis was carried out in triplicate.

#### 2.3.3. Sensory Evaluation

A hedonic scale test was used to measure the color of instant powder, as well as the color, flavor, and overall enjoyableness of the drink. Powder samples were placed on white dishes. Drinks were placed in white cups. Each of the consumers received three samples and was asked to rate them based on the degree of liking them on a nine-point hedonic scale (1 = dislike extremely, 5 = neither like nor dislike, 9 = like extremely). All samples were identified with random three-digit numbers, and the consumers were not informed of the ingredients. Consumers were provided with warm water to rinse after each sample. The evaluation was conducted in a classroom with white light, a temperature of 25 ± 1 °C, and a relative humidity of 60 ± 5% RH. Untrained consumers (*n* = 90, 19–56 years old, 37 males and 53 females) were recruited from the students, staff, and faculty of Hungkuang University.

### 2.4. Statistical Analysis

Each measurement was conducted in triplicate, except for the hedonic test (n = 90). The experimental data were subjected to an analysis of variance using the Statistical Analysis System software package version 9.4 (SAS Institute, Cary, NC, USA). When a significant difference was found among treatments, Duncan’s multiple range test was used to determine the differences among the mean values at the level of α = 0.05. 

## 3. Results and Discussion

### 3.1. Quality Characteristics of Blanched and Centrifuged Broths

The washed shiitake (2.5 kg) was placed in boiling water (12.5 kg) and blanched for 5 min. After blanching five batches of shiitake with the same boiling water, the BB obtained after filtration weighed 12.04 kg, and the yield was 96.32% (Table 1). After centrifuging the thawed shiitake, the CB obtained after filtration comprised 404.7 g/kg of blanched shiitake, and the yield was 40.47%. The above results suggest that much of the BB and CB recovered from the processing process represents a heavy burden for sewage treatment. The moisture (%), soluble solids (°Brix), and pH values of the BB and CB were 99.20 and 97.67, 0.90 and 2.90, and 6.14 and 6.40, respectively (Table 1). On a dry basis, the ash and protein levels were significantly higher in freeze-dried BB than in freeze-dried CB, while crude fat and carbohydrate showed the reverse (Table 1). The dry matter of BB and CB contained 16.88–19.21% of protein, 12.89–13.50% of ash, and 67.28–70.00% of carbohydrate. The protein, fat, and ash of the shiitake leak into the blanching water to a higher degree than carbohydrate during hot blanching process [5]. Indeed, when comparing our results with the proximate compositions of fresh shiitake described in Mau et al. [5], the ratios of protein and ash in BB and CB are here increased, while carbohydrates show the reverse. 

Mushrooms contain large amounts of functional compounds, and these components remain in their by-products. Therefore, the reuse of mushroom by-products has been widely discussed [14,37]. The solids also contain substantial amounts of bioactive components, such as crude polysaccharides, ergothioneine, and total phenols (Table 1). However, γ-aminobutyric acid was not detected in either broth, and ergosterol was not detected in the BB, while the CB contained 0.01 g of ergosterol per 100 g dry matter. Further, when comparing the bioactive components between BB and CB, the content of crude polysaccharides in BB was obviously higher than that in CB, whereas the contents of ergothioneine and total phenols in CB were significantly higher than in BB.

Amongst these bioactive components, polysaccharides are considered the most frequently recycled component, due to their health benefits and high presence in mushroom by-products [14]. Ultrasound was used to extract the polysaccharides from the by-products generated from *Agaricus bisporus*, and the highest extraction yield was 4.7% [38]. The contents of polysaccharides in BB and CB were 7.45% and 4.53%, respectively. Ergothioneine was found at high levels in mushrooms and demonstrated a high antioxidant ability. The dry matter of *L. edodes* contained 0.92 mg/g of ergothioneine [39]. The ergothioneine contents of BB and CB were calculated at 1.28 and 1.92 mg/g dry matter, respectively. The total phenolic content of the water extract of *L. edodes* and *Volvariella volvacea* showed 1.33 and 1.34 mg of GAEs/g of dry mushroom, which suggests higher phenol contents and antioxidant activities compared to the methanol extract [40]. The total phenols of BB and CB were 0.152 and 0.362 mg GAE/100 g dry matter, respectively. According to the above results, the blanched and centrifuged broths maintained a large amount of polysaccharides, ergothioneine, and total phenols when compared to the fruiting bodies of the mushrooms. These components are important for maintaining health, and so BB and CB show a high reuse value. The freeze-dried solids of BB and CB contained high amounts of non-volatile taste components, including sugars and polyols (40.56–45.69%), free amino acids (6.58–6.69%), and 5′-nucleotides (0.98–1.47%) (Table 1). The sugars and polyols were significantly higher in CB than in freeze-dried BB, while 5′-nucleotides showed the reverse.

The main sugars and polyols in both freeze-dried broths were mannitol (33.155–34.315 g/100 dry matter), arabitol (4.684–7.476 g/100 dry matter), and trehalose (2.715–3.888 g/100 dry matter) (Table 2). The total sugar and polyol content of freeze-dried CB was significantly higher than that of BB. In terms of taste-related components, arabinose, fructose, and glucose were not detected in either of the freeze-dried broths. There was no significant difference in total free amino acid content between the two freeze-dried broths (Table 3). The main amino acids in the two freeze-dried broths were glutamine, followed by arginine and glutamic acid. All essential amino acids (EAA) were found in both freeze-dried broths, and all non-essential amino acids (NEAA) were detected, except cystine. Except for the higher content of phenylalanine in freeze-dried BB, there was no significant difference in EAA between the freeze-dried broths. Of the NEAAs, the aspartic acid, glutamic acid, glycine, and proline levels were significantly higher in the BB than in the CB, while the glutamine level was significantly higher in the CB than in the BB, and the rest of the NEAAs showed no significant differences between the two freeze-dried broths. There were no significant differences in the concentrations of total amino acids (TAA), EAAs, NEAAs, and branched-chain amino acids (BCAAs) in the two freeze-dried broths. The MSG-like amino acids in BB were significantly more prevalent than in CB, while sweet amino acids showed the reverse. The bitter and tasteless amino acids did not differ significantly between the two freeze-dried broths. In both freeze-dried broths, the highest 5′-nucleotide was 5′-CMP (Table 4). The BB was significantly higher in all 5′-nucleotides than in the CB, except for 5′-UMP, which showed no significant difference. The contents of flavor 5′-nucleotides were 1.97-fold higher in BB than in CB and 1.50-fold higher in total 5′-nucleotides. The sugars and polyols showed lower molecular weights and higher water solubility compared to 5′-nucleotides. This may explain the CB maintaining a higher ratio of arabitol, mannitol, and trehalose but a lower ratio of 5′-nucleotides after centrifugation.

Using the addition equation established via sensory evaluation by Yamaguchi et al. [30], the EUC values of the BB and CB were found to be 858.06 and 309.66 g MSG/100 g dry matter, respectively (Table 1). The EUC was significantly higher in BB than in CB. Compared with CB, the EUC of BB is 2.77-fold higher. EUC values can be grouped into four levels: (1) >1000 g MSG/100 g, (2) 100–1000 g MSG/100 g, (3) 10–100 g MSG/100 g, and (4) <10 g MSG/100 g [41]. The BB and CB both reached level 3, indicating that our blanched and centrifuged broths were rich sources of umami compounds.

Potassium, magnesium, sodium, calcium, phosphorus, zinc, manganese, selenium, iron, and copper are essential minerals for the human body and play important roles in human cell metabolism, biosynthesis, and physiological functions [2]. The blanching process damages the cell structure, causing the release of minerals into the blanching water [5]. Indeed, BB and CB contain a variety of essential mineral elements (Table 5). The most prevalent major mineral in both freeze-dried broths was potassium (3387.6–3708.2 mg/100 g dry matter), followed by magnesium (183.2–230.2 mg/100 g dry matter) and sodium (129.5–132.3 mg/100 g dry matter). Potassium intake can reduce the risk of cardiovascular and renal disease. However, most populations around the world consume less than the recommended amount of potassium, whereas the intake of sodium is generally double the recommended amount [42,43]. Moreover, the BB and CB samples were shown to be high in potassium and low in sodium. Potassium can reduce the absorption of sodium and has been used as a low-sodium salt. It could act as a good food source for people with high blood pressure. Due to their advantage of having lower sodium, the mushrooms were mixed into a beef taco filling to reduce the overall sodium [6].

In regard to the trace minerals, both BB and CB had the highest amounts of zinc (1.811–7.042 mg/100 g dry matter), followed by manganese (0.479–3.26 mg/100 g dry matter) and selenium (1.811–7.042 mg/100 g dry matter). In particular, the CB contained more manganese and zinc than BB, while selenium was the reverse. The trace minerals are important to maintaining human health. In particular, selenium enhances some of the biological activities of mushrooms [44]. As such, blanched and centrifuged broths are good sources of essential mineral elements.

### 3.2. Quality Characteristics of Instant Powders

The concentrated broth (BBC and CBC) was visually darker than the original broth (BB and CB), and the BBC was visually darker than the CBC (Appendix A). In order to produce an instant drink powder with more health benefits, FS-2 was utilized as a carrier. Figure 1 shows the color of spray-dried instant powders prepared from mixtures of BBC or CBC with FS-2. The BB instant powder was yellower than the CB powder. The moisture contents of the BB and CB instant powders (4.41–7.06%, 1.72–3.55%) were, in descending order, SD13 > SD14 > SD15 (Appendix A). As the level of FS-2 increased, the water activity (0.315–0.365, 0.226–0.307) also decreased significantly. This finding is related to changes in the moisture level in instant powder. It seems that FS-2 could retain less water and reduce the water activity, which means that the shelf-life can be extended. Among the three ratios of FS-2, 1:5 is recommended, because a higher ratio of microcapsule packaging helps to reduce the risk of core components interacting with the external environment.

In BB and CB instant powders, the *L** and *WI* values increased significantly with the increase in FS-2, while the *a** (except CB) and *b** values decreased significantly (Appendix A), which means that with the increase in FS-2, the instant powder became brighter, whiter, less red, and less yellow. The CB instant powder was brighter, whiter, less red, and less yellow than the BB instant powder. These two instant powders were less bright, less white, redder (except CB), and yellower than FS-2, because the instant powders contained BB or CB. The colors of the BB and CB instant powders were compared with FS-2, and their Δ*E* values decreased with the increase in FS-2, a difference which could even be observed with the naked eye. Song et al. [45] also reported an Δ*E* > 3 and a color difference that was obvious to the human eye. The CB instant powders were whiter than the BB instant powder, and their color was closer to that of FS-2. However, according to the sensory evaluation, the color of the powder was insignificant, and the scores were around 7.4–7.6.

### 3.3. Sensory Evaluation of Instant Drink

All instant drinks were made by brewing instant powder with boiling water. Figure 2 shows the color of the instant drinks in which instant powders were brewed in boiling water. As the solid concentrations of BB or CB increased, the drinks became darker. For all instant drinks, as the solid concentration of BB or CB increased, the *L** and *h** values decreased significantly, while *a*, b*, c*,* and Δ*E* showed the reverse (Appendix A). The *h** values of all instant drinks were 80.77–98.50° (Appendix A), which implies that they were between yellow (60°) and green (120°) [46], as is consistent with the color of the instant drinks shown in Figure 2. 

Table 6 shows the results of the hedonic sensory evaluation of instant drinks. As the concentration of BB in the drinks increased, the color scores were lower, but the highest color score of the CB instant drink was 2.0–2.5%. The flavor and overall preference scores of all instant drinks increased from 1.0% to 2.5% and then decreased from 2.5% to 3.5%, indicating that the drink with 2.5% instant powder was the most popular. Consumers believed that in terms of the flavor intensity, 1.0% to 2.0% was too weak, but more than 3.0% was too strong and had a slightly bitter taste; 2.5% was the best. The flavor and overall preference were similar between BB and CB. It seems that the significant differences in taste components between BB and CB, such as in sugars and polyols, 5′-nucleotides, EUC, and MSG-like, and sweet amino acids, did not affect the hedonic sensory results. 

A heatmap based on the correlation between color parameters and the hedonic sensory scores of the instant drinks has been constructed (Figure 3). The color scores of the hedonic sensory evaluation show high positive correlations between *L** and *h (°)** and negative ones between *a**, *b*,* and *c**. Interestingly, the BB drink showed stronger relations when compared to CB. This indicates that the dark color of BB impacts its color hedonic score more strongly than in the case of CB.

## 4. Conclusions

Blanched and centrifuged shiitake broths are rich in polysaccharides, total phenols, ergothioneine, essential minerals, and taste compounds and are low in fat and sodium. This indicates that the blanching water of shiitake is not only tasty but also has health benefits. The BB and CB were made into functional instant drinks using functional FS-2 as a carrier, which was dried via spray-drying. The color became whiter as more FS-2 was added. However, in the sensory evaluation, the color of the powder was insignificant. The solid contents of BB and CB in the instant drinks caused significant differences in the degrees to which the consumers liked them, with 2.5% solid content being the most popular. According to the data obtained in this study, when BB and CB were mixed with FS-2 to produce an instant drink, it will not only be more acceptable for the consumers but will also offer greater health benefits. We suggest that manufacturers should use functional FS-2 to make the blanched and centrifuged broths produced during the production of vacuum-fried shiitake crisps into functional instant drinks, which can not only reduce wastewater treatment costs but also increase economic benefit.

## Figures and Tables

**Figure 1 foods-12-02925-f001:**
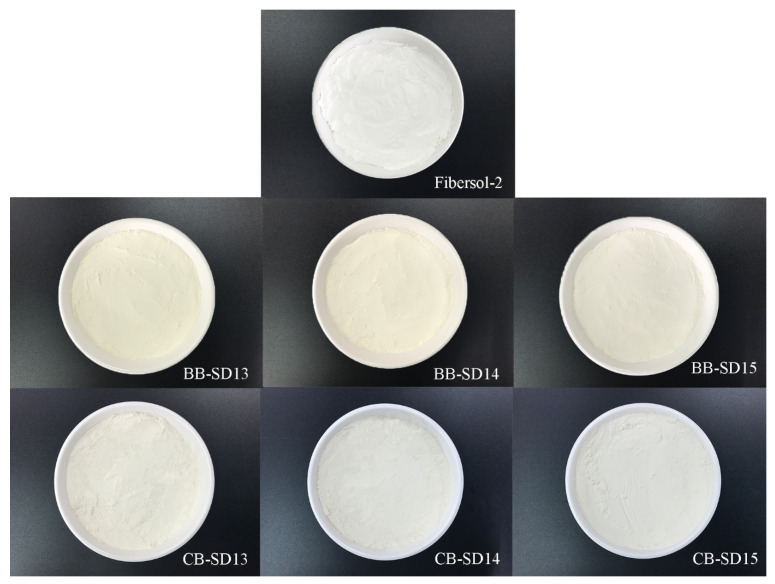
The appearance color of Fibersol-2, BB, and CB instant powders.

**Figure 2 foods-12-02925-f002:**
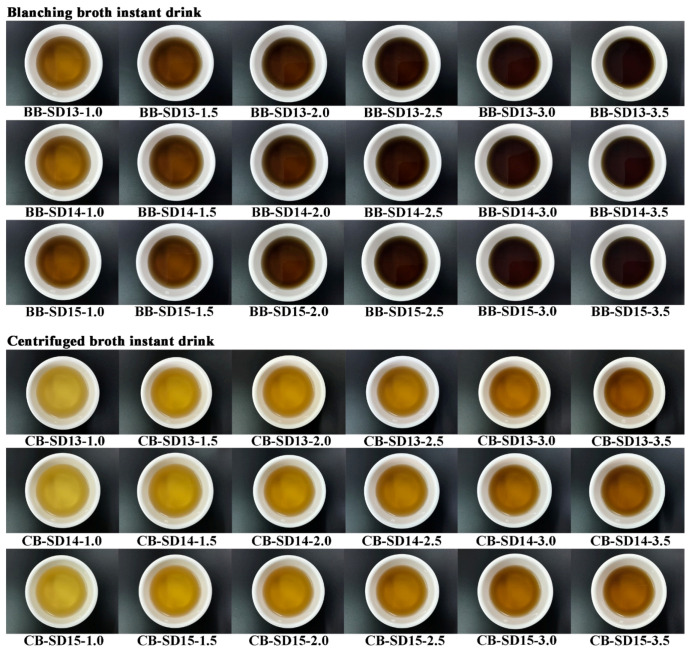
The appearance color of instant drinks.

**Figure 3 foods-12-02925-f003:**
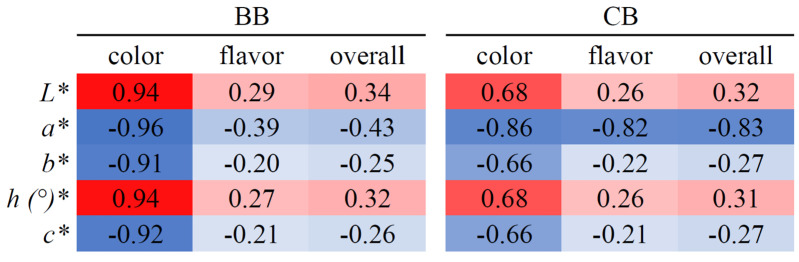
The correlation heatmap of instant drink and color property.

**Table 1 foods-12-02925-t001:** The physicochemical quality characteristics of blanched and centrifuged broths from shiitake.

	BB	CB
Yield (%) ^1^	96.32 ± 0.63 ^3^	40.47 ± 0.76
Moisture (g/100 g)	99.20 ± 0.04	97.67 ± 0.01
Soluble solids (°Brix)	0.90 ± 0.10	2.90 ± 0.10
pH	6.14 ± 0.04	6.40 ± 0.06
Proximate composition (g/100 g dry matter)		
Crude protein	19.21 ± 0.73 ^a^	16.88 ± 0.04 ^b^
Crude fat	0.01 ± <0.01 ^b^	0.23 ± 0.01 ^a^
Crude ash	13.50 ± 0.06 ^a^	12.89 ± 0.05 ^b^
Carbohydrate	67.28 ± 0.77 ^b^	70.00 ± 0.45 ^a^
Bioactive composition (g/100 g dry matter)		
γ-Aminobutyric acid	nd	nd
Crude polysaccharide	7.45 ± 0.36 ^a^	4.53 ± 0.17 ^b^
Ergosterol	nd ^4^	0.01 ± <0.01
Ergothioneine	0.19 ± <0.01 ^b^	0.19 ± <0.01 ^a^
Total phenols ^2^	0.15 ± 0.01 ^b^	0.36 ± 0.02 ^a^
Taste composition (g/100 g dry matter)		
Sugars and polyols	40.56 ± 0.86 ^b^	45.69 ± 1.01 ^a^
Free amino acids	6.58 ± 0.11 ^a^	6.69 ± 0.04 ^a^
5′-Nucleotides	1.47 ± 0.02 ^a^	0.98 ± 0.02 ^b^
EUC (g MSG/100 g sample)	858.06 ± 7.63 ^a^	309.66 ± 39.15 ^b^

^1^ BB yield (%) = (filtered weight of blanched broth after blanching 5 batches of shiitake/weight of initial blanching water) × 100. CB yield (%) = (filtered weight of centrifuged broth/weight of blanched shiitake) × 100. ^2^ Unit: expressed as mg GAE (gallic acid equivalent)/100 g dry matter. ^3^ Each value is expressed as a mean ± standard deviation (*n* = 3). Values with different lowercase letters within a row differ significantly (*p* < 0.05). ^4^ nd: not detected.

**Table 2 foods-12-02925-t002:** Sugars and polyols of blanched and centrifuged broths from shiitake.

	Content (g/100 g Dry Matter)	
	BB	CB
Arabinose	nd ^1^	nd
Arabitol	4.684 ± 0.306 ^b,2^	7.476 ± 0.188 ^a^
Fructose	nd	nd
Glucose	nd	nd
Mannitol	33.155 ± 0.447 ^b^	34.315 ± 0.804 ^a^
Trehalose	2.715 ± 0.178 ^b^	3.888 ± 0.203 ^a^
Total	40.554 ± 0.862 ^b^	45.679 ± 1.008 ^a^

^1^ nd: Not detected. ^2^ Each value is expressed as mean ± standard deviation (*n* = 3). Values with different lowercase letters within a row differ significantly (*p* < 0.05).

**Table 3 foods-12-02925-t003:** Free amino acids of blanched and centrifuged broths from shiitake.

Amino Acids	Content (g/100 g Dry Matter)	
	BB	CB
Essential amino acid		
Histidine	0.099 ± 0.006 ^a,2^	0.109 ± 0.010 ^a^
Isoleucine	0.142 ± 0.002 ^a^	0.134 ± 0.003 ^a^
Leucine	0.241 ± 0.006 ^a^	0.222 ± 0.006 ^a^
Lysine	0.276 ± 0.017 ^a^	0.279 ± 0.015 ^a^
Methionine	0.294 ± 0.022 ^a^	0.280 ± 0.006 ^a^
Phenylalanine	0.135 ± 0.002 ^a^	0.122 ± 0.001 ^b^
Threonine	0.231 ± 0.012 ^a^	0.237 ± 0.023 ^a^
Tryptophan	0.031 ± 0.001 ^a^	0.028 ± 0.003 ^a^
Valine	0.305 ± 0.007 ^a^	0.299 ± 0.003 ^a^
Non-essential amino acid		
Alanine	0.521 ± 0.020 ^a^	0.457 ± 0.008 ^a^
Arginine	0.803 ± 0.040 ^a^	0.834 ± 0.019 ^a^
Asparagine	0.198 ± 0.005 ^a^	0.184 ± 0.003 ^a^
Aspartic acid	0.215 ± 0.004 ^a^	0.175 ± 0.004 ^b^
Cystine	nd ^3^	nd
Glutamic acid	0.730 ± 0.017 ^a^	0.638 ± 0.016 ^b^
Glutamine	1.676 ± 0.014 ^b^	2.110 ± 0.042 ^a^
Glycine	0.256 ± 0.013 ^a^	0.190 ± 0.012 ^b^
Proline	0.093 ± 0.001 ^a^	0.074 ± 0.001 ^b^
Serine	0.232 ± 0.006 ^a^	0.213 ± 0.003 ^a^
Tyrosine	0.105 ± 0.008 ^a^	0.103 ± 0.008 ^a^
Total amino acids (TAA)	6.583 ± 0.112 ^a^	6.688 ± 0.042 ^a^
Essential amino acids (EAA)	1.754 ± 0.001 ^a^	1.710 ± 0.034 ^a^
Non-essential amino acids (NEAA)	4.829 ± 0.111 ^a^	4.978 ± 0.076 ^a^
Branched-chain amino acid (BCAA) ^1^	0.688 ± 0.014 ^a^	0.655 ± 0.011 ^a^
Taste characteristic		
MSG-like	0.945 ± 0.022 ^a^	0.813 ± 0.020 ^b^
Sweet	3.009 ± 0.040 ^b^	3.281 ± 0.018 ^a^
Bitter	2.050 ± 0.071 ^a^	2.028 ± 0.023 ^a^
Tasteless	0.381 ± 0.025 ^a^	0.382 ± 0.023 ^a^

^1^ Branched-chain amino acid = leucine + isoleucine + valine. MSG-like (monosodium glutamate-like) = aspartic acid + glutamic acid. Sweet amino acids = alanine + glutamine + glycine + proline + serine + threonine. Bitter = arginine + histidine + isoleucine + leucine + methionine + phenylalanine + tryptophan + valine. Tasteless = lysine + tyrosine. ^2^ Each value is expressed as mean ± standard deviation (*n* = 3). Values with different lowercase letters within a row differ significantly (*p* < 0.05). ^3^ nd: Not detected.

**Table 4 foods-12-02925-t004:** 5′-Nucleotides of blanched and centrifuged broths from shiitake.

	Content (g/100 g Dry Matter)	
	BB	CB
5′-AMP ^1^	0.179 ± 0.012 ^a,3^	0.128 ± 0.001 ^b^
5′-CMP	0.684 ± 0.012 ^a^	0.486 ± 0.002 ^b^
5′-GMP	0.383 ± 0.008 ^a^	0.151 ± 0.017 ^b^
5′-IMP	0.018 ± 0.002 ^a^	0.007 ± 0.001 ^b^
5′-UMP	0.182 ± 0.002 ^a^	0.190 ± 0.003 ^a^
5′-XMP	0.022 ± 0.001 ^a^	0.019 ± 0.001 ^b^
Flavor 5′-Nucleotides ^2^	0.602 ± 0.006 ^a^	0.305 ± 0.017 ^b^
Total 5′-Nucleotides	1.468 ± 0.020 ^a^	0.981 ± 0.022 ^b^

^1^ 5′-AMP, 5′-adenosine monophosphate; 5′-CMP, 5′-cytidine monophosphate; 5′-GMP, 5′-guanosine monophosphate; 5′-IMP, 5′-inosine monophosphate; 5′-UMP, 5′-uridine monophosphate; 5′-XMP, 5′-xanthosine monophosphate. ^2^ Flavor 5′-Nucleotides = 5′-IMP + 5′-GMP + 5′-XMP + 5′-AMP. ^3^ Each value is expressed as mean ± standard deviation (*n* = 3). Values with different lowercase letters within a row are significantly different (*p* < 0.05).

**Table 5 foods-12-02925-t005:** Minerals of blanched and centrifuged broths from shiitake.

Mineral	Content (mg/100 g Dry Matter)	
	BB	CB
Major mineral		
Calcium (Ca)	56.7 ± 2.3 ^a,1^	33.7 ± 1.5 ^b^
Magnesium (Mg)	183.2 ± 7.7 ^b^	230.2 ± 3.0 ^a^
Phosphorus (P)	2.0 ± 0.1 ^b^	16.1 ± 0.5 ^a^
Potassium (K)	3387.6 ± 59.8 ^b^	3708.2 ± 12.3 ^a^
Sodium (Na)	132.3 ± 4.9 ^a^	129.5 ± 4.1 ^b^
Trace mineral		
Aluminum (Al)	0.117 ± 0.005 ^a^	0.080 ± <0.001 ^b^
Chromium (Cr)	0.008 ± <0.001 ^b^	0.029 ± 0.001 ^a^
Iron (Fe)	0.285 ± 0.013 ^a^	0.168 ± 0.005 ^b^
Manganese (Mn)	0.479 ± 0.016 ^b^	3.260 ± 0.132 ^a^
Selenium (Se)	0.474 ± 0.016 ^a^	0.055 ± 0.001 ^b^
Zinc (Zn)	1.811 ± 0.082 ^b^	7.042 ± 0.342 ^a^

^1^ Each value is expressed as mean ± standard deviation (*n* = 3). Values with different lowercase letters within a row differ significantly (*p* < 0.05).

**Table 6 foods-12-02925-t006:** Hedonic sensory evaluation of instant drinks prepared from instant drink powders with hot water.

	BB			CB		
	Color	Flavor	Overall	Color	Flavor	Overall
SD13-1.0 ^1^	7.2 ± 0.9 ^A,2^	4.5 ± 0.9 ^E^	4.4 ± 0.6 ^F^	6.9 ± 0.8 ^E^	4.4 ± 0.9 ^E^	4.3 ± 0.6 ^F^
SD13-1.5	7.1 ± 0.9 ^A^	5.5 ± 1.1 ^D^	5.4 ± 0.8 ^E^	6.8 ± 0.7 ^E^	5.4 ± 1.0 ^D^	5.3 ± 0.8 ^E^
SD13-2.0	6.6 ± 1.0 ^B^	6.6 ± 0.8 ^C^	6.8 ± 0.8 ^B^	7.4 ± 0.8 ^ABC^	6.7 ± 0.8 ^C^	6.9 ± 0.7 ^B^
SD13-2.5	6.5 ± 1.0 ^B^	7.3 ± 0.8 ^A^	7.3 ± 0.8 ^A^	7.3 ± 0.9 ^ABC^	7.4 ± 0.7 ^A^	7.5 ± 0.9 ^A^
SD13-3.0	5.5 ± 1.1 ^CD^	4.3 ± 0.5 ^EF^	3.9 ± 0.7 ^G^	6.0 ± 0.8 ^F^	4.3 ± 0.5 ^EF^	3.9 ± 0.7 ^HI^
SD13-3.5	5.3 ± 1.0 ^CD^	3.7 ± 0.5 ^H^	3.3 ± 0.5 ^IJ^	5.9 ± 0.8 ^F^	3.8 ± 0.6 ^H^	3.4 ± 0.8 ^K^
SD14-1.0	7.1 ± 0.7 ^A^	4.4 ± 1.4 ^E^	4.4 ± 1.4 ^F^	7.0 ± 0.7 ^E^	4.1 ± 0.8 ^G^	4.1 ± 0.7 ^GH^
SD14-1.5	7.4 ± 1.0 ^A^	5.6 ± 1.0 ^D^	5.2 ± 0.7 ^E^	7.3 ± 1.0 ^CD^	5.7 ± 1.0 ^D^	5.2 ± 0.8 ^E^
SD14-2.0	6.4 ± 1.3 ^B^	6.7 ± 1.1 ^C^	6.4 ± 1.0 ^C^	7.4 ± 0.9 ^ABC^	6.9 ± 1.0 ^BC^	6.9 ± 1.0 ^C^
SD14-2.5	6.6 ± 1.4 ^B^	7.1 ± 0.9 ^A^	7.3 ± 1.0 ^A^	7.6 ± 0.7 ^A^	7.2 ± 0.9 ^AB^	7.3 ± 0.8 ^A^
SD14-3.0	5.6 ± 0.8 ^C^	4.0 ± 0.9 ^G^	3.7 ± 0.9 ^GH^	5.8 ± 0.7 ^F^	4.0 ± 1.0 ^G^	3.7 ± 1.0 ^IJ^
SD14-3.5	5.3 ± 0.5 ^CD^	3.4 ± 0.6 ^I^	3.2 ± 0.9 ^J^	5.5 ± 0.7 ^G^	3.4 ± 0.7 ^I^	3.1 ± 0.6 ^L^
SD15-1.0	7.3 ± 1.1 ^A^	4.2 ± 0.8 ^EFG^	4.4 ± 0.7 ^F^	7.0 ± 0.9 ^E^	4.1 ± 0.8 ^FG^	4.3 ± 0.6 ^FG^
SD15-1.5	7.3 ± 1.0 ^A^	5.7 ± 1.1 ^D^	5.6 ± 0.7 ^D^	7.1 ± 1.0 ^DE^	5.6 ± 1.1 ^D^	5.6 ± 0.6 ^D^
SD15-2.0	6.7 ± 1.0 ^B^	6.8 ± 1.2 ^BC^	6.6 ± 0.7 ^BC^	7.3 ± 0.8 ^BCD^	6.9 ± 1.2 ^BC^	6.7 ± 0.5 ^BC^
SD15-2.5	6.4 ± 1.3 ^B^	7.0 ± 1.1 ^AB^	7.1 ± 1.2 ^A^	7.6 ± 1.0 ^AB^	7.3 ± 1.2 ^A^	7.3 ± 1.1 ^A^
SD15-3.0	5.3 ± 0.5 ^CD^	4.0 ± 0.6 ^FG^	3.5 ± 0.9 ^HI^	5.4 ± 0.6 ^G^	4.1 ± 0.9 ^FG^	3.6 ± 0.9 ^JK^
SD15-3.5	5.2 ± 1.0 ^D^	3.2 ± 0.9 ^I^	2.8 ± 0.5 ^K^	5.3 ± 1.1 ^G^	3.3 ± 1.2 ^I^	2.9 ± 0.6 ^L^

^1^ SD: Spray drying. 13, 14, and 15: dry solid of blanched broth (or centrifuged broth) concentrate/indigestible dextrin (*w*/*w*). 1.0, 1.5, 2.0, 2.5, 3.0, and 3.5: instant drinks containing 1.0%, 1.5%, 2.0%, 2.5%, 3.0%, and 3.5% dried broth, respectively. ^2^ Each value is expressed as mean ± standard deviation (*n* = 90). Means with different capital letters within a column differ significantly (*p* < 0.05). Nine-point hedonic scale with 1, 5, and 9 representing extreme dislike, neither like nor dislike, and extreme like, respectively.

## Data Availability

All available data are contained within the article.

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
