# Peer review of "Quality Evaluation of Shiitake Blanched and Centrifuged Broths as Functional Instant Drinks"

_foods, 2023, doi:10.3390/foods12152925_

Round 1

Reviewer 1 Report

The authors report the use of shiitake blanched and centrifuged broths for instant drink. The manuscript is well-written, and all discussion are supported by experimental data and references. Moreover, the methods are described in detail. Below are my considerations to improve the quality of the manuscript:

1. Is there any material characterization of the Fibersol-2, BB and CB instant powders.  it will help to understand the materials rather than just showing the real photo of materials. 

2. Did the author performed the test about how long does this BB and CB instant drink retain its color and flavors?

3. Can this method to prepare instant drink to be applied to other type of mushrooms? why the author chooses only shiitake? 

4. The references used in this work are relevant, however, there are too many references are old. please use up to date references. 

Reviewer 2 Report

Abstract

- Use of this term “consumer’s preference” (L#22) is more appropriate when they were asked to compare a sensory attribute between samples.

Introduction

- L#57-60 serves as rationale for this study. Please provide some literature review of similar studies.

- L#76-77, please elaborate in a broader context. Are there any other carrier with similar function to FS-2?

- L#79-79 please check grammar and outline “few studies”.

Materials and Methods

- On 2.2.1 Preparation section, please elaborate on food safety issue.

Results and Discussion

- L#220-221, please elaborate in detail.

Conclusions

- L#482-484, is this conclusion based on authors’ own speculation or the evidence from this study? Functional benefit of both broths should be highlight in both qualitative and quantitative term.

some gramma errors.

Reviewer 3 Report

Just a kind suggestion (no need to address this comment) that authors may use a CIE1931 figure to visually present the color change of the powder.
